# Identification of Marine Biotechnology Value Chains with High Potential in the Northern Mediterranean Region

**DOI:** 10.3390/md21070416

**Published:** 2023-07-22

**Authors:** Ana Rotter, Antonia Giannakourou, Jesús E. Argente García, Grazia Marina Quero, Charlène Auregan, George Triantaphyllidis, Amalia Venetsanopoulou, Roberta De Carolis, Chrysa Efstratiou, Marina Aboal, María Ángeles Esteban Abad, Ernesta Grigalionyte-Bembič, Yannis Kotzamanis, Mate Kovač, Maja Ljubić Čmelar, Gian Marco Luna, Cristóbal Aguilera, Francisco Gabriel Acién Fernández, Juan Luis Gómez Pinchetti, Sonia Manzo, Iva Milašinčić, Antun Nadarmija, Luisa Parrella, Massimiliano Pinat, Efstratios Roussos, Colin Ruel, Elisabetta Salvatori, Francisco Javier Sánchez Vázquez, María Semitiel García, Antonio F. Skarmeta Gómez, Jan Ulčar, Cristian Chiavetta

**Affiliations:** 1Marine Biology Station Piran, National Institute of Biology, Fornače 41, 6330 Piran, Slovenia; ernesta.grigalionyte-bembic@nib.si (E.G.-B.); jan.ulcar@nib.si (J.U.); 2Institute of Oceanography, Hellenic Centre for Marine Research, 46.7 km Athens-Sounio Avenue, 19013 Anavyssos, Greece; agiannak@hcmr.gr (A.G.); avenetsanopoulou@hcmr.gr (A.V.); ch.efstra@hcmr.gr (C.E.); 3Department of Information and Communication Engineering, University of Murcia, Avda. Teniente Flomesta, 30003 Murcia, Spain; jesus.argente@um.es (J.E.A.G.); skarmeta@um.es (A.F.S.G.); 4CNR IRBIM, National Research Council—Institute of Marine Biological Resources and Biotechnologies, Largo Fiera della Pesca, 60125 Ancona, Italy; grazia.quero@irbim.cnr.it (G.M.Q.); direttore@irbim.cnr.it (G.M.L.); massimiliano.pinat@irbim.cnr.it (M.P.); 5Pôle Mer Méditerranée, Toulon Var Technologies, 93 Forum de la Méditerranée, 83190 Ollioules, France; auregan.charlene@outlook.fr (C.A.); ruel@polemermediterranee.com (C.R.); 6Laboratory of Fish Nutrition and Omics Technologies, Institute of Marine Biology, Biotechnology and Aquaculture, Hellenic Centre for Marine Research, Iera Odos 86, 11855 Athens, Greece; gvtrianta@hcmr.gr (G.T.); jokotz@hcmr.gr (Y.K.); e.roussos@hcmr.gr (E.R.); 7Department for Sustainability, ENEA (Italian National Agency for New Technologies, Energy and Sustainable Economic Development), Via Anguillarese, 301, 00196 Rome, Italy; roberta.decarolis@enea.it (R.D.C.); sonia.manzo@enea.it (S.M.); luisa.parrella@enea.it (L.P.); elisabetta.salvatori@enea.it (E.S.); cristian.chiavetta@enea.it (C.C.); 8Department of Plant Biology, Faculty of Biology, University of Murcia, Avda. Teniente Flomesta, 30003 Murcia, Spain; maboal@um.es; 9Department of Cell Biology and Histology, Faculty of Biology, University of Murcia, Avda. Teniente Flomesta, 30003 Murcia, Spain; aesteban@um.es; 10Croatian Agency for SMEs, Innovations and Investments—HAMAG-BICRO, Ksaver 208, 10000 Zagreb, Croatia; mate.kovac@hamagbicro.hr (M.K.); maja.ljubiccmelar@hamagbicro.hr (M.L.Č.); iva.milasincic@hamagbicro.hr (I.M.); antun.nadramija@hamagbicro.hr (A.N.); 11Institute of Agri Food Research and Technology, Crta. Poble Nou 5.5 km, 43540 La Ràpita, Spain; cristobal.aguilera@irta.cat; 12Department of Chemical Engineering, University of Almeria CIESOL, Cañada San Urbano, 04120 Almeria, Spain; facien@ual.es; 13Spanish Bank of Algae, Institute of Oceanography and Global Change, University of Las Palmas de Gran Canaria, Muelle de Taliarte, 35214 Telde, Spain; juan.gomez@ulpgc.es; 14Department of Physiology, Faculty of Biology, Regional Campus of International Excellence “Campus Mare Nostrum”, University of Murcia, Avda. Teniente Flomesta, 30003 Murcia, Spain; javisan@um.es; 15Department of Applied Economics, University of Murcia, Avda. Teniente Flomesta, 30003 Murcia, Spain; mariase@um.es

**Keywords:** marine biotechnology, blue biotechnology, innovation, value chains, Northern Mediterranean, microalgae, macroalgae, IMTA, circular economy, discards valorization

## Abstract

Marine (blue) biotechnology is an emerging field enabling the valorization of new products and processes with massive potential for innovation and economic growth. In the Mediterranean region, this innovation potential is not exploited as well as in other European regions due to a lack of a clear identification of the different value chains and the high fragmentation of business innovation initiatives. As a result, several opportunities to create an innovative society are being missed. To address this problem, eight Northern Mediterranean countries (Croatia, France, Greece, Italy, Montenegro, Portugal, Slovenia and Spain) established five national blue biotechnology hubs to identify and address the bottlenecks that prevent the development of marine biotechnology in the region. Following a three-step approach (1. Analysis: setting the scene; 2. Transfer: identification of promising value chains; 3. Capitalization: community creation), we identified the three value chains that are most promising for the Northern Mediterranean region: algae production for added-value compounds, integrated multi-trophic aquaculture (IMTA) and valorization aquaculture/fisheries/processing by-products, unavoidable/unwanted catches and discards. The potential for the development and the technical and non-technical skills that are necessary to advance in this exciting field were identified through several stakeholder events which provided valuable insight and feedback that should be addressed for marine biotechnology in the Northern Mediterranean region to reach its full potential.

## 1. Introduction

To cope with an increasing global population, the rapid depletion of many resources, increasing environmental pressures and climate change, and as a consequence of the social and cultural changes that are taking place in consumers, Europe needs to radically change its approach to the production, consumption, processing, storage, recycling and disposal of biological resources [1]. Therefore, it is necessary to shift the focus to novel sources to develop new products and processes. In this regard, the marine environment, which is vast, largely underexplored and unexploited, represents the opportunity to valorize marine resources [2]. This is carried out through marine biotechnology, a sector that explores the marine bio-resources (marine organisms and ecosystems) as potential sources of innovation and a major factor of economic growth, thus representing a significant contributor to the blue bioeconomy. This is also endorsed by the European Commission that is supporting the development of marine-related economic and innovation activities, in particular since the adoption of the Communication on Blue Growth in 2011 when the blue economy was adopted as a central element of the European Union’s (EU) Integrated Maritime Policy in implementing the Europe 2020 strategy for sustainable and inclusive growth [3]. Recently, the European Commission has published a communication aimed at integrating ocean policy into Europe’s new economic policy to ensure that the ‘blue economy’ plays a major role in the implementation of the European Green Deal (EGD), clearly stating that the dualism between environmental protection and the economy must be overcome [4]. The biotechnology sector in the EU directly and indirectly creates one million jobs, has a gross added value of EUR 78.7 billion and is expected to grow globally by at least 50% by 2030 [5,6]. Marine biotechnology solutions, coupled with circular economy business models, are excellent tools for closing production cycles and ensuring resource-efficient processes, avoiding waste and keeping resources in use for as long as possible through cascading biomass use and recycling, generating added value products in numerous contexts while ensuring, at the same time, the protection of natural capital in the ocean.

The Mediterranean Basin, located across the South of Europe and the North of Africa, is unique by virtue of its history, cultural heritage climate, diet and ecosystems, and by it being a global hotspot of biological diversity (with over 17,000 species) with a high rate of endemism [7,8,9]. The Mediterranean Basin is mostly oligotrophic and its biological productivity decreases from north to south and west to east, while the opposite trend is observed for salinity and temperature [10]. In theory, the Mediterranean thus offers natural and societal opportunities to valorize the rich biodiversity and advance in the marine biotechnology and the blue bioeconomy sectors. However, in practice, the Mediterranean region has been showing a slower economic growth and employment performance than other middle-income countries, mostly linked to its poor business environment, complicated, ambiguous and often excessive bureaucracy, inadequate production systems, competitiveness and technological transfers [11,12]. This is unfortunate, as Southern European countries (including the Northern Mediterranean region) have excellent education systems and research expertise, but relatively modestly funded research bases, lower investment into infrastructures and few and/or fragmented bio-businesses [6]. Moreover, there is Mediterranean fragmentation at a regional/national level (in policies, legislation and business initiatives). As marine biotechnology is a relatively young discipline, its key players are not clearly identified at (inter)national levels and the lack of coordination on the key enablers limits its huge growth potential in the region. Therefore, the creation of national, regional and international collaborative networks can establish critical mass, provide a platform for knowledge transfer, facilitate political and financial support and produce the most creative and innovative results to address important societal challenges [6,13]. Significantly, the establishment of collaborative efforts necessarily relies on teamwork, with a transdisciplinary combination of experts with technical and non-technical skills, such as social, communication, technology transfer, entrepreneurship, legal, intellectual property protection, business strategy and market research, among others [14,15].

With this in mind, the B-Blue project—“Building the blue biotechnology community in the Mediterranean”, financed through the Interreg Mediterranean Programme between September 2020 and September 2022 (https://b-blue.interreg-med.eu/, accessed on 8 May 2023), triggered the creation of a collaborative network, composed of 10 partners from 8 Northern Mediterranean countries (Croatia, Greece, France, Italy, Montenegro, Portugal, Slovenia and Spain, see Figure 1) aiming to identify promising marine biotechnology value chains and thus unlock the innovation potential of the region. These eight countries differ in their maturity level of marine biotechnology, which was recently assessed in terms of these indicators: (i) aquatic macroorganism aquaculture, macro- and microalgal aquaculture; (ii) existence of marine (blue) biotechnology in national and regional Smart Specialization Strategies (Table 1); and (iii) obtained funding for marine biotechnology projects and support measures [3]. By combining these indicators, four groups are represented in this study: Mediterranean leaders (France and Spain), with implemented innovation strategies, national/regional legislation and existing industry; followers (Italy and Portugal), with existing legislative and financial support but several value chains—see Table 1—that were not fully finalized; emerging countries (Greece and Slovenia), with some legislative support but a currently lower financial support and less established aquaculture sector; and newcomers (Croatia and Montenegro), with established commitment on scientific or legislative side but lacking concrete implemented innovations.

Through a series of events, organized in the eight Mediterranean countries within the frame of the B-Blue project, valuable interactions and feedback were sought from relevant and transversal stakeholders from the region, resulting in the identification of the most relevant value chains (see Table 1) to be addressed in each territory. The remainder of the article presents the three-step strategy and the results obtained to highlight their potential to assist the growth of marine biotechnology in the Northern Mediterranean region in the future.

## 2. Results

To develop the most promising marine biotechnology value chains (Table 1) for the Northen Mediterranean region, project partners conducted the work in a three-step process: analyze-transfer-capitalize. (1) The analysis step consisted of jointly agreeing on the requirements that are most important for advancing the marine biotechnology sector. (2) The transfer step enabled the identification of the most promising marine biotechnology value chains for the Northern Mediterranean region. (3) The capitalization step provided tools to establish national blue biotechnology hubs.

### 2.1. Step 1: Analysis. Setting the Scene: Fundamental Requirements to Advance in Marine Biotechnology in the Northern Mediterranean Region

Out of the 81 potential activities that were identified by the Blue Bioeconomy Forum [18] as important for advancing in the field of marine biotechnology, three were most often selected by expert respondents—project partners from eight Northern Mediterranean countries—as being the most important. These can be represented as three specific action points: legislation, financing, and collaboration through knowledge creation. 

### 2.2. Step 2: Transfer. Identification of Most Promising Value Chains in the Northern Mediterranean Region

We first identified the existing good practices in the marine biotechnology sector that are in development, already implemented or ready to enter the market. A total of 89 good practices involved at least one of the project partners’ countries (the Northern Mediterranean countries of Croatia, Greece, France, Italy, Montenegro, Portugal, Slovenia and Spain). These individual good practices focused on one, more or all sectors (aquaculture, cosmetics, energy, environment, feed industry, industrial processes—enzyme and catalysts, nutraceuticals, pharmaceuticals and other) and are shown in Figure 2. Among these, 55 good practices deal with a specific group of organisms (Figure 3), while the rest are either of a strategic nature, or are not specific for any particular type of organism. The full list of good practices is available as Appendix A.

### 2.3. Step 3: Capitalization. Creation of the Community by Establishment of Blue Biotechnology Hubs

To capitalize the existing knowledge and prevent its loss that typically occurs after the end of financing rounds, we first conducted a stakeholder mapping exercise (see Materials and Methods on how to conduct and map the stakeholders, which optimizes the engagement effort to yield new collaborations, funding or legislation change). A total of 636 potentially interested stakeholders from several sectors and activities (administration, research and academia, industry and small/medium enterprises (SMEs), non-governmental organizations (NGOs), past and current projects, media) from eight Mediterranean countries (Croatia, Greece, France, Italy, Montenegro, Portugal, Slovenia and Spain) were identified. These stakeholders are important to establish national communities of practice. We followed the living labs approach that enables the co-creation of innovative user-oriented solutions. The creation of living labs was selected as they provide a resource for collective innovation for new products, processes, and services [19]. We shall call them “Blue Biotechnology Hubs (BBt Hubs)”. Five selected Northern Mediterranean territories established these hubs—France, Greece, Italy, Slovenia and Spain. They were designed to be open to different stakeholders to allow them to co-create new knowledge and solutions. Through several types of setups (almost 40 events organized: workshops, networking events, work cafés, hackathons and strategic meetings), we addressed over 1500 individuals that are directly or indirectly involved in, or can contribute to, the developing marine biotechnology value chains in the Northern Mediterranean region.

## 3. Discussion

We followed a three-step approach to define the value chains with most potential in the Northern Mediterranean region: (1) analysis, (2) transfer and (3) capitalization. By analyzing the existing initiatives and good practice examples, we first created the knowledge repository on a national level. This also enabled the identification/confirmation of the bottlenecks in the Mediterranean region (as presented in the Introduction section): a lack of identified experts and the high fragmentation of activities, where research and innovation are often developed as isolated initiatives without coordination activities, which could ease access to the market. We continue with the discussion on each of the three steps and detail our approach to identify, create and capitalize knowledge on the value chains with most potential in the Northern Mediterranean region.

### 3.1. Step 1: Analysis—Identification of Bottlenecks

The identified bottlenecks that call for specific requirements to set the scene for advancing innovation in the field of marine biotechnology in the Northern Mediterranean region can broadly be clustered in three categories: legislation and policy support, financing and collaboration through knowledge creation.

#### 3.1.1. Policy Support

*European policies*. Within the EU, the blue economy is expected to play a major role in the long-term strategy to reach the full transformation toward sustainable growth stated by the EGD, that will build on targets such as carbon neutrality, circular economy, zero pollution and the restoration of biodiversity [20]. The Mediterranean Sea, being a strategic crossroad for the history, economy and culture of European, Middle Eastern and North African countries, will be central for this transformation [21]. When dealing with seafood, for example, wild fisheries, today representing the main source of food from the oceans, have made, in recent years in the EU, considerable efforts to bring fish stocks back to sustainable levels and to reduce overfishing to meet the Common Fisheries Policy and its standards. Aquaculture has the potential to be an alternative source of sustainable seafood and to further become a large source of low-impact food. However, the positioning of mariculture as a seafood fix as compared to wild-capture fisheries still has to be fully established [22], in light of the numerous challenges to be addressed in order to also fully meet the SDGs and to operate under a One Health Perspective [23]. Aquaculture will need to develop more research and innovation including the testing of new sustainable sources of food and feed, the developing of new marketing standards and the promotion of decarbonization and circular economy practices, while facilitating its coexistence with other sectors of the blue economy. In this sense, the recently launched large-scale investment by Horizon Europe in research (EU Mission on Oceans and Waters “Restore our Ocean and Waters by 2030”) including its Partnership “A climate neutral, sustainable and productive Blue Economy” (in brief “Sustainable Blue Economy Partnership” or SBEP), the new European Maritime, Fisheries and Aquaculture Fund (EMFAF) and others will certainly boost the innovation and advancing of this sector, and foster the transformative change required to achieve a transition of the blue economy that will benefit the planet, its people and the economy. Furthermore, the Strategic EU Blue Economy partnership supports actions towards the ocean dimensions of sustainable development in the context of the UN 2030 Agenda with its three pillars, namely sustainability, climate neutrality and productivity.

*Trans-Regional policies*. Following the Conference on “Strengthening Euro-Mediterranean Cooperation through Research and Innovation” held in La Valletta in 2017, under the auspices of the Maltese Presidency of the Council of the EU, La Valletta Declaration was released. The Declaration restates the belief that strengthening Euro-Mediterranean cooperation in research and innovation contributes to fully tapping the potential of economic growth and sustainable development of the Mediterranean region and commits to knowledge creation and identification of innovative solutions. In addition, the BLUEMED initiative (http://www.bluemed-initiative.eu/, accessed on 17 March 2023) has been a lighthouse in the Mediterranean region, aiming to advance a shared vision for a healthier, more productive, resilient, better known and valued Mediterranean Sea, promoting the citizens’ social well-being and prosperity, and boosting economic growth and jobs. The BLUEMED Strategic Research and Innovation Agenda, which was updated in 2018, outlines a set of key challenges for the Mediterranean region including different sectors of interest, such as ecosystems, climate change, biotechnologies, aquaculture, fisheries, observing systems, offshore platforms and spatial planning.

*National policies*. The smart specialization strategy (S3, Table 1) has presented a profound structural revolution in the way innovation policies are conceived by identifying local potentials, local needs and advocating between investments in knowledge and human capital, and the present industrial and technological ‘vocations’ and competences of territories [24]. It facilitates regional and national diversified and decentralized specialization into areas that should secure existing and future competitiveness, thus pivoting to regional innovation [25,26]. Indeed, the S3 should prioritize domains, areas and economic activities where regions or countries have a competitive advantage or have the potential to generate knowledge-driven growth and to bring about the economic transformation needed to tackle the major and most urgent challenges for the society and the natural and built environment. Importantly, although an initial set of priorities is identified during the first design of the S3, they can be changed or modified when new information/developments makes it advisable. Hence, it comes to no surprise that several Mediterranean experts identified the possibility of participating in the national S3 as a window of opportunity for providing feedback and positioning themselves as national experts, resource and knowledge providers, thus creating actual impact and changing the policy content. For example, during the period of the project implementation, the national S3 for most Mediterranean countries were in their upgrade period, thus providing the opportunity to include sustainable blue economy as an important horizontal area that is well integrated to other national priority areas.

#### 3.1.2. Increase Funding at National Level

Once the policy support is achieved, other pressing issues for bringing the marine biotechnology (blue) products to the market are addressed, for example: increased funding at the national level to support the continuation of the development of marine biotechnology products and processes, also through engagement of the industry sector in research and development. Nationally, measures need to be developed to incentivize researchers/companies to collaborate in blue economy sectors. These funding tools can be split into two main categories. Firstly, the public support for investments in industrial research and development (R&D) activities, which then also include collaboration between universities and industry, including SMEs, and secondly, matching funds for non-research investments in marine biotechnology sector to advance the non-technical skills. Ideally, national funding would critically review the capitalization potential of the concluded activities and only those with realistic prospects to further increase their technological readiness would be financed in the next rounds, thus preventing the loss of knowledge and results generated during the financing rounds.

#### 3.1.3. Education, Training, Mentoring and Coaching

Besides reaching a policy consensus and enabling financial support, it is imperative to provide adequate opportunities for human capital development. This can be carried out on two levels, either before (i.e., during formal education) or after entering the workforce. There are several useful activities for knowledge development in the field of marine biotechnology. (i) Firstly, designing mentorship/coaching activities on technical and non-technical skills, where skilled individuals (coaches, mentors) support clients (at any career level) by providing formal and informal training and guidance. Coaching is more focused on performance and specific tasks or objectives, while mentoring provides a more generalized support. Establishing mentoring or coaching programs provide benefits to mentors, protégés and organizations, but not all organizations have such programs in place [27]. When these coaching activities are coupled with networking opportunities, the results can yield increases in all stages of the marine biotechnology value chain, from knowledge acquisition and participation in innovative collaboration networks to economic benefits [13,28]. (ii) Secondly, providing start-ups in the sector with advice on business and financing. An example are business incubators that offer either training, support or workspace. Another example are seed accelerators, typically targeting existing companies and/or products or processes that are in the higher stages of technology readiness. (iii) Thirdly, developing targeted workshops and trainings on cooperation between the blue bioeconomy sector and universities. (iv) Furthermore, assisting local marine biotechnology producers: for example, creating taste labs, school campaigns and education classes. (v) And finally, designing Mediterranean-based international study programs on bachelor’s and PhD levels with scholarships. Good practice examples in the Mediterranean region include two projects. The first one is DEEP BLUE (Developing Education and Employment Partnerships for a Sustainable Blue Growth in the Western Mediterranean Region) [29], which was financed by the European Maritime and Fisheries Fund (EMFF). The second project is the BlueSkills (Blue Jobs and Responsible Growth in the Mediterranean) [30] regional project, endorsed by the Union for the Mediterranean and aiming to promote blue employment by developing skills, exchanging knowledge and valorizing research for a more sustainable Mediterranean Sea. However, since these good practices depend on the financing scheme and are time-constricted by the duration of individual projects, the sustainability and longevity of marine biotechnology/bioeconomy (blue) education in the Mediterranean region needs to be considered in the near future, possibly by iterating the current national educational curricula, especially the tertiary level ones that provide most in-depth specialization opportunities.

### 3.2. Step 2: Transfer—Promising Value Chains in the Northern Mediterranean Region

After assessing the Northern Mediterranean good practices benchmarking exercise (Figure 2 and Figure 3) we defined the value chains that are currently best represented in the Northern Mediterranean region. Algae (macro- and microalgae) production to obtain added-value compounds, represented in over 50% of the target organisms (Figure 3), and aquaculture or fisheries discard valorization, that was represented in around 25% of the good practices sectors (Figure 2), were on top of the good practices lists. We therefore focus on these sectors as the ones with the highest potential for advancing the innovation status in the Northern Mediterranean region. We further categorized the aquaculture sector (Figure 2) into two emerging value chains: IMTA, and the valorization of aquaculture/fisheries/processing by-products Category 3, unavoidable/unwanted catches and discards. Taken together with algal biotechnology (Figure 3), these are the three most promising value chains with high relevance for the selected countries in the Northern Mediterranean region. They are discussed in the subparagraphs below.

#### 3.2.1. Algae Production for Added-Value Compounds

In the ambitious strategy for developing bioeconomy in Europe, algae represent an emerging biological resource of great importance for its potential applications in different fields. They have biosorption potential, can sequester atmospheric CO_2_, provide biofuel, feed and food supplements, food ingredients, nutraceuticals and cosmetics, among others [14].

In the Northern Mediterranean countries (Portugal included) there are currently 280 companies producing algae found through European Marine Observation and Data Network—EMODnet [31] and through the project partners from Mediterranean countries. The produced organisms by companies from different Mediterranean countries within the B-Blue project are shown in Figure 4. The figure includes the genus *Arthrospira* (*Spirulina*) as well, as this cyanobacterium is widely used for industrial production due to their biocompounds content, including biopeptides, biopolymers, pigments, carbohydrates, essential fatty acids, minerals, oligoelements and sterols [32]. The taxonomic classification of *Arthrospira* (*Spirulina*) is an ongoing discussion in the scientific community [33,34] and is beyond the scope of this work, hence, it is represented as a standalone taxon which is also historically very important from the consumers’ and the industrial perspective [34,35]. Their widespread production is also clearly seen in the Northern Mediterranean region, with France owning the majority (72%) of companies specialized in the cultivation of this cyanobacterium. Microalgae and macroalgae have fewer producer companies. Interestingly, macroalgae are produced in only three of the B-Blue Mediterranean countries (France, Spain and Portugal). The official data have no records about microalgae cultivation in Greece, Slovenia and Croatia. However, Greece has proven experience and capacity in microalgae cultivation of the species *Isochrysis* sp., *Chlorella* sp., *Tetraselmis* sp., *Nannochloropsis* sp., *Rhodomonas* sp. and *Dunaliella* sp. on an industrial scale, as part of the daily routine of several hatcheries, which support larvae feeding in fish farms. Croatia has an enterprise in microalgae cultivation (hatchery) and Slovenia reports three enterprises in the last five years producing small quantities of microalgae and *Spirulina*.

##### Macroalgal Production

The global production of macroalgae amounts up to 35 million tons (Mt) of fresh weight annually (where 97% of its biomass derives from aquaculture and the rest is harvested) and around 24% and less than 1% of it is contributed within the EU for harvesting and aquaculture, respectively [36]. The global production is still primarily dominated by two Asian countries, namely China and Indonesia, producing together >90% of the global market supply [37]. Hence, although European production is currently small-scale, the macroalgal sector is considered as a notable subsector in European blue bioeconomy and the projections imply an expansion in European annual production from around 0.3 up to 8 Mt by 2030 which could create up to 85,000 jobs [20,38]. It is estimated that the key seaweeds grown in the Northern Mediterranean countries (including their Atlantic coasts) in 2030 could include, for instance, sea lettuce (*Ulva lactuca*) for human consumption, sugar kelp (*Saccharina latissima*) for use in food products and animal feed, dulse (*Palmaria palmata*) for the food and cosmetics sectors, *Asparagopsis taxiformis* for cattle feed additives with methane-reducing properties, and oarweed (*Laminaria digitata*) to produce alginate for use in the food additives and biopackaging segments [38]. An emerging field is also seaweed farming for the ecological restoration of marine macroalgal forests, given the predicted upscaling from small-scale, short-term academic experiments to industry and restoration practitioners, required to secure the oceans’ sustainable future [39,40,41] and building up in the Mediterranean region on the knowledge produced by successful projects such as AFRIMED (http://afrimed-project.eu/, accessed on 17 March 2023) and ROCPOP-life (http://www.rocpoplife.eu/, accessed on 17 March 2023). Regardless on the target value chain, however, it is important to use the basic ecological knowledge of these species and avoid their introduction into new areas, especially where they might be considered invasive, such as is the case for some *Asparagopsis* species [42]. Macroalgal species which are mostly produced in B-Blue countries are presented in Figure 5. Although macroalgae production at a commercial scale is recorded only in three B-Blue countries, other B-Blue countries (such as Greece, Italy and Slovenia) are cultivating macroalgae at a pilot scale as part of several research programs or projects (aquaculture in open tanks and small aquariums). Many edible macroalgae have high nutritional values and the environmentally friendly cultivation and harvesting methods make them appealing as raw material for food and feed [43,44,45].

Over 70% of registered macroalgal production in Mediterranean countries is attributed to harvesting, mostly manual, which faces the challenge of balancing the socio-economic and environmental sustainability of the activity [35]. It is therefore of strategic importance to promote macroalgal aquaculture, which on one hand presents a higher potential for scaling up the production volumes, while on the other one being associated with high infrastructural and operational costs [35]. Seaweed farming is often considered as the least environmentally damaging form of aquaculture and has been shown to have positive effects on many ecosystem services [46]. The species commercially harvested are primarily *Laminaria hyperborea*, *Ascophyllum nodosum*, *Fucus* sp., *Himanthalia elongata*, *Porphyra* sp. and *Undaria* sp. [47], while *Saccharina latissima*, *Laminaria* sp., *Palmaria palmata*, *Chondrus crispus* and *Ulva* sp. are the species mainly cultivated [43]. Seaweed production (both harvesting from wild stocks and aquaculture) is primarily concentrated in the Atlantic region with few units of cultivating species that are native in the Mediterranean Sea e.g., *Ulva* sp. and *Gracilaria* sp. [35]. The main reasons behind the slower adoption of macroalgal cultivation in the Mediterranean waters are the needs to: (i) target native Mediterranean species with adapted cultivation techniques for specific sectors/purposes/locations/species; (ii) create germplasm banks to ensure the preservation of desirable local traits and genetic diversity; (iii) ensure the availability of suitable cultivation sites; but also (iv) a lack of investment; and (v) the restrictive and inflexible implementation of European environmental regulations on aquaculture [48,49,50]. Mechanical harvesting is undertaken by boats and is mainly practiced in France (Brittany), Spain (Galicia and Asturias) and, to a lesser degree, in the Basque country (France). Manual harvesting of macroalgae and gathering of storm cast macroalgae are important in France, Spain and Portugal [51]. Harvesters either gather the cast or cut macroalgae at low tide. Diving is another way to harvest macroalgae manually and is practiced mostly in Portugal [51]. Macroalgae are cultivated in land-based tanks or ponds or in sea-based (coastal and offshore) structures such as long-lines or rafts [52]. They can be cultivated as a monoculture or IMTA, which is further discussed below. However, the quantities of cultivated European macroalgae are insufficient in volume, too expensive and produced by a fragmented supply chain, concentrated in the Atlantic area. This represents an opportunity for the Northern Mediterranean region to promote the development of this exciting sector in the region, providing that careful attention is placed also on the environmental factors (such as the difference in water temperature between the Atlantic and the Mediterranean waters). Currently, no Mediterranean species is being grown and the efforts should be focused on the search of autochthonous species (by definition perfectly adapted to the environmental conditions). So far only a few companies have managed to secure a license for large-scale operations and leverage sufficient funding to expand. The demand for “all-natural ingredients” has been on the rise, owing to the safety concerns associated with synthetic ingredients; hence, the demand for macroalgal protein-based products is expected to grow considerably in the coming years.

##### Microalgal Production

The microalgal species (freshwater and marine species) that are most often cultivated in the B-Blue countries are presented in Figure 6. Some are used in the food industry, such as the green algae *Chlorella* sp., which is internationally identified as “Generally Recognized As Safe” (GRAS), a certification legislated under the United States Food and Drug Administration—FDA [53]. Some species of these microalgae are also included under the EU Novel food regulation. Other certified GRAS species include the green algae *Haematococcus* sp. and *Dunaliella* sp. [53]. Microalgae are also used in the aquaculture feed sector and as feed ingredients for livestock [54,55]. When introducing algae in aquafeeds, the results show an improvement in the health of the fish due to the probiotic compounds contained into the microalgal biomass [56]. The microalgal biotechnology industry is of high interest due to their pigments, carbohydrates, lipids and proteins productivity potential that can be used as food and feed supplements and a broad variety of other applications, such as pharmaceuticals, cosmetics or agriculture ones [57,58]. However, the bulk production of microalgae carbohydrates and proteins for the food and feed sector has not yet grown in accordance with its potential, because it requires higher production volumes and consequently the boosting of the cost-effective scale-up [58,59,60]. Moreover, the prices are not yet comparable with cheaper land-based alternatives, such as soy protein and palm oil. Nevertheless, the development of technology keeps boosting the development of cost and technology optimizations for microalgal applications. Microalgae are cultivated using different production systems. (i) The photo-bioreactors, commonly (71%) used for microalgae production in Europe, are generally capital intensive, allowing a stricter control of the environmental factors and biomass quality as well as an increase in the photosynthetic efficiency and productivity [61,62]. (ii) Open ponds entail a lower investment but have a high risk of contamination, lower control of the environmental conditions and greater land and water requirements [63,64]. (iii) Finally, fermenters refer only to heterotrophic algae. Large-scale cultivation will decisively contribute to the development of a sustainable industry for biomass production as well as generate cost-effective high-value products. Indeed, the development of suitable and cost-effective biomass processing and harvesting strategies that contribute to around 30% of the final biomass cost, is still ongoing [57,65].

#### 3.2.2. Integrated Multi-Trophic Aquaculture—IMTA

Animal aquaculture is one of the fastest growing food production sectors that has been generating concerning consequences for the environment, including chemical and biological pollution, disease outbreaks, unsustainable feeds and competition for coastal space [66,67]. To promote sustainable approaches, IMTA systems combine fed aquaculture species with inorganic extractive species (e.g., macroalgae) and organic extractive species (e.g., suspension- and deposit-feeders) cultivated in proximity [68]. Finfish, macroalgae, mollusks, crustaceans, sea cucumbers, sea urchins, sponges, polychaetes and plants (such as *Salicornia europaea* and *Aster tripolium*) have all been identified to have a high potential for inclusion in efficient IMTA [69,70,71,72,73]. Through IMTA, some of the uneaten feed and wastes, nutrients, and by-products, considered “lost” from the fed component, are recaptured and converted into harvestable and healthy seafood of commercial value [74]. These systems significantly contribute to the sustainability of aquaculture through economic, societal and environmental benefits, including the recycling of waste nutrients from higher trophic-level species into the production of lower trophic-level species, thus increasing production efficiency, product and economic diversification and social acceptability of farmed products [68,69]. In addition, biomass produced in IMTA systems, including those from macroalgae, bivalve shells or other types of waste, can be further exploited for other (blue and green) biotechnological applications. Hence, the development of IMTA value chains can significantly contribute to the establishment of circularity in aquaculture where waste streams from one industry provide the raw materials for another. At the same time, IMTA has the potential to reduce the nutrients and organic matter inputs from finfish aquaculture, thus having a significant environmental impact on fish farms, while creating appropriate business models to increase profitability. Furthermore, this concept aligns with recommendations made in the Food from the Oceans report (2017), which highlighted the need to expand low- and multi-trophic marine aquaculture as an ecologically efficient source of increasing food and feed [75].

The efforts to implement IMTA on a large scale are mainly concentrated in the Atlantic area and North Europe and only trials carried out on an experimental/laboratory scale exist in the Northern Mediterranean region [69,71,73]. Currently, IMTA has several important challenges. There is a need to address and estimate the potential for contaminants in uneaten feed and feces to bioaccumulate by organic extractive and inorganic extractive species, potentially reentering the food chain supply [76]. Similarly, current legislations still appear inadequate to cover the co-cultivation of multiple species in proximity [76]. Hence, there are no general rules and guidance for practitioners, which is one of the main factors for the lack of trust by stakeholders in the field in adopting IMTA on an industrial level. Different culture environments affecting the growth of extractive species lead to various bio-mitigation capacities. As most of the land-based experiments are conducted under controlled conditions, interactions between co-cultured species and their natural physical and ecological environments cannot be well represented [77].

Mediterranean coasts are densely populated and there is a high level of competition for coastal space utilization. Nevertheless, fish and mollusk aquaculture has already been established in the Northern Mediterranean region and in the B-Blue countries therein. Indeed, Eurostat reports that France, Greece and Spain are the top producers of European seabass and seabream (Greece and Spain jointly having 75% and 71% of total EU production of seabass and seabream, respectively) [78]. France and Spain are the top producers of mussels in the EU, jointly having 63% of the total EU production [78]. Therefore, the well-established aquaculture sector offers suitable preconditions for developing large-scale cultivation and the development of IMTA systems.

On the other hand, for each species cultured, different markets exist with different demands, potentials and constraints, all of which add to the likelihood of increasing costs before revenues are generated [79]. However, although the share of aquafarming in Europe has not increased significantly since 2008 and has maintained a constant 20% share of the total fisheries production, its production value has doubled and represents 41% of the total value of the EU’s total production of fishery products in 2017 [78]. This share can even be dramatically increased if implementing IMTA practices, considering their potential ecological efficiency drive, environmental acceptability, product diversity, profitability and benefit for society [80]. From a more long-term perspective, IMTA has the potential to provide ecosystem services and benefits not only at the farm level but rather at broader environmental and societal levels. Additionally, consumer awareness should not be excluded in the case of IMTA. Along the seafood value chain, the way Mediterranean aquaculture economies are organized may change, with consumers asking for sustainably farmed seafood from traceable and transparent sources, and aquaculture offering “on-demand” sustainable products from selective and safe farms.

IMTA can be well incorporated into the industrial symbiosis concept, a process where waste/by-products from one industrial level represent resources for the next industrial process. IMTA entails the establishment of a collaborative web of knowledge, businesses, material and energy exchanges among different players in the industrial symbiosis networks [81]. Industrial symbiosis demands collaborations offered by geographic proximity [82]. This concept is, therefore, of high relevance to increase the economic prosperity of Northern Mediterranean countries as the business opportunities diversify and multiply by creating additional value chains through IMTA.

#### 3.2.3. Aquaculture/Fisheries/Processing By-Products Category 3, Discards and Bycatch Valorization in Added Value Sectors

Discards, or discarded catch, include target species or any other species that constitute a part of the catch during fishing operations but are not retained on board and are returned at sea dead or alive [83]. In the Mediterranean bottom trawl fisheries, the discard rate is estimated at 13.3–26.8%, with an average of 18.6% [84,85]. The mitigation of discards is a major concern to conservation bodies and the wider public, as discards are linked to mortality of juvenile fishes, benthic species and biodiversity loss [86]. Bycatch refers to species, with or without commercial value, that are not targeted by the fishery [83]. It is estimated that bycatch represents around 40% of global marine catches [84]. Finally, the by-products result from the processing of fishery and aquaculture products in the commercial and processing chain, including gutting, scaling, skinning, filleting, gill removal and head removal. The by-products (e.g., heads, bones, guts, shells, among others) represent between 30 and 70% of the whole fish and often go unutilized, ending up as waste [87]. The estimations for discards and by-products for the eight Mediterranean countries are provided in Table 2 and Table 3, respectively, and amount to a range of between 1,069,422 and 2,142,859 tons yearly. These wastes represent a major ecological and economical issue [88]. On the other hand, they are also an opportunity, where a promising solution is their valorization as raw materials for the production of high-added-value compounds in the pharmaceutical, feed, food, energy, cosmetic industry and in agriculture, instead of being discarded [89,90]. Fishery and aquaculture by-products are indeed extremely rich in high-quality compounds (such as proteins having different biological properties), leading to the great potential for the bioprocess industry to exploit these highly valuable products. Indeed, landfilling along with other municipal waste is the last option and should not be considered as a valid option [90]. It is projected that large amounts of new fish biomass will be generated in European ports following the Landing Obligation Guidelines issued by the EU [91,92]. Despite having several good practices in Northern Europe [89,90], aquaculture/fisheries/processing by-products, unavoidable/unwanted catches and discards in the Mediterranean Sea are currently without doubt an underutilized sector.

#### 3.2.4. Identified Specific Bottlenecks for Promising Northern Mediterranean Value Chains

Besides the general bottlenecks that are preventing the advancement of marine biotechnology activities in the Northern Mediterranean region, as discussed in Step 1: Analysis, specific bottlenecks have been identified for each of the three promising value chains (Figure 7). These have been identified through rounds of consultations during B-Blue activities—workshops, trainings, work cafés, networking events, hackathons and strategic meetings. Importantly, the list of the specific bottlenecks is a living document. The bottlenecks might change over time, depending on the financial, research, regulatory and time efforts spent to develop the three value chains.

Concerning the development of algae production, there is a need to improve data quality and additional research, knowledge sharing and collective action to expand cultivation beyond the currently limited commercial species. Cooperation among stakeholders, collaboration with producers and research institutions should be integrated into R&D to define good practices and introduce cost-effective technologies for algal producers. To optimize the time efficiency and cost-effectiveness of the production of algal biomass and compounds, further market research is needed to understand the opportunities of the sector in the Northern Mediterranean region. This can lead to proposing strategies to manage social and economic risks. Importantly, the standardization of biomass production, licensing framework and licensing procedures are needed, as they can affect processing and production capabilities. Current processing technologies allow the obtaining of a large portfolio of products for largely different applications. However, to increase the relevance of the algal sector, additional end products must be targeted, which requires the development of new processing technologies. Additionally, the import of products with misleading labelling and abiding by specific EU/national directives (e.g., irradiation, heavy metals, etc.) is an issue that has to be properly addressed. In terms of licensing and patenting, these steps may significantly delay the time to market due to the lengthy compliance processes, on one hand, and the patent regulation obstacles on the other, considering also that patent protection is time limited and has to be renewed after several years [93]. Moreover, patent subjects cannot be published until the patent is approved and even after the patent is granted, the protection is territory- and time-limited [93]. Finally, adequate infrastructure is needed for algae production. This will not be feasible without public incentives. However, incentives to promote national production are limited, including the quality control of imported biomass, which often represents unfair competition between producers and importers. Interestingly, an opportunity to develop this emerging value chain has resulted as a consequence of the COVID-19 pandemic, that is, the need to use local resources [94,95]. It is therefore no longer advisable to buy biomass from abroad or even overseas, but rather to use local suppliers. This is also driven by the market and end-users looking for high value-added products. In the Mediterranean region, the extensive coastline offers suitable preconditions for developing large-scale cultivation of algal biomass. Several production technologies have already been piloted, however, a bottleneck in the economically viable exploitation of macro- and microalgae is the lack of an efficient, industrial scale cultivation technology. The establishment of regional pilot plants and small biorefineries could encourage and boost further investments. The expansion of microalgae farming, large on-shore or off-shore seaweed farming, and especially innovative applications, require extensive support in both capital investment and, most likely, in their short-medium term operation until profit sea-margin can be achieved. Few success stories exist worldwide and they all started up with significant access to research and capital credit, either public or public-private partnerships, until they can be financially self-sustainable on their revenues.

To advance the development of IMTA, a continuous R&D is essential to understand the biological, biochemical, hydrographic, oceanographic and seasonal processes, as is: the suitable selection of species, considering the warm oligotrophic Mediterranean waters; the adaption and development of new technologies; addressing the engineering, operational protocol, and economics of these technologies; and a model development flexible and friendly enough so that they can be tailored and adjusted to the specifics of each particular site. There is a scarcity of adequate infrastructures for IMTA production. Hence, the competitiveness with introduced products will not be feasible without public incentives. Moreover, technology should reach maturity and prove economic viability through the pilot-scale/demonstration sites before it can be commercially implemented. The results from these demonstrations must be coupled with profound market research. Financial benefits for the aquaculture companies are one of the most important factors to guarantee the industrial support and willingness to implement IMTA. In the Northern Mediterranean region that is heavily dependent on tourism, awareness raising campaigns should target the potential consumers and industry to understand the environmental benefits and eliminate the stigma of IMTA being a “waste of coastal space”, as was observed by the stakeholders. Funding IMTA systems as nature-based solutions for environmental remediation supporting ecosystem goods and services is a challenge. The scalability of IMTA, i.e., the ability to start growing new species at a small scale and investing lots of time, money, and other resources before the implementation of large-scale infrastructure, demands for testing these systems under several operational workflow combinations before their implementation. Finally, limitations to the implementation of IMTA at a commercial scale are derived at the national level of regulations (as being one of the general bottlenecks for advancement of marine biotechnology value chains), coupled with the fundamental aspects of authorization and licensing, environmental impact assessment and the co-cultivation of different species.

One of the main challenges for the valorization of discards and fish by-products is the willingness of the fishing industry (fisheries, aquaculture and processing) to maintain continuous R&D for the valorization of discards and by-products in the B-Blue countries. In turn, the scientific sector must increase their efforts to promote effective collaborations with the industrial sector that yield tangible results. Currently however, infrastructures that are able to handle fish by-products produced by the catching sector are limited. Moreover, there is a need to develop business models. These include the collection, handling, maintenance and transfer of raw material, the creation of practical solutions for the separation and proper storage and the development/adaptation of protocols for production/extraction of added-value compounds. These business models support a circular economy approach with the potential of connecting blue and green economies. In the broader Mediterranean region, no successful solutions are applied yet, and tons of unexplored biomass ends up in the sea or in landfills. One of the main challenges for the valorization of discards and fish by-products is the available quantities for processing in the Mediterranean region: we estimated from slightly over 1 Mt (optimistically low estimation) to slightly over 2.1 Mt of discards and by-products yearly in the eight Northern Mediterranean countries (Table 2 and Table 3). The complex logistics behind novel supply chains need to be developed in order to collect and distribute these wastes so they are available for further processing. The final identified bottleneck that prevents the implementation of waste valorization value chains in the Northern Mediterranean region is the necessity to raise awareness and change the perception on waste to be rather considered as a resource in the eyes of consumers as well as fishermen and aquaculture experts. Indeed, any solution for the valorization of waste must involve the local fisheries communities that should be educated and trained for the environmental and financial viability of such initiatives.

### 3.3. Step 3: Capitalization—Creation of National Mediterranean Blue Biotechnology Hubs

The creation of national Mediterranean blue biotechnology hubs enabled the addressing of the state-of-the-art on the national level, the analysis of encountered challenges and opportunities, the anticipation of emerging issues and the identification of bottlenecks for development and needs/priorities for the different stages of the value chains to advance the innovation potential in the Northern Mediterranean region. Moreover, the identified stakeholders from the stakeholder mapping (from scientific, academic, industrial, governmental and other public or nongovernmental sectors) were contacted and had the opportunity to promote themselves, their expertise and their potential involvement in the identified value chains with the most potential. The non-technical development of value chains consisted of several activities. Almost 40 activities were organized within the B-Blue project’s lifetime, which attracted over 1500 participants. This high interest in marine biotechnology strongly suggests the need for networking opportunities in the Northern Mediterranean region to advance the innovation potential. The selection of activities depended on the local context and state of development of the value chain within each blue biotechnology hub. The activities conducted (national workshops, work cafés, networking events, technical workshops, hackathons and strategic meetings) are detailed in the Materials and Methods section. Importantly, the establishment of blue biotechnology hubs enables the creation of localized resource and activity centers. This can effectively result in setting up new collaborative agendas for research and development that will help finalize the technology readiness levels, form regulatory guidelines, advocate for policy change, put on market new products and processes, contribute to local economy, and thus increase the innovation level in the Northern Mediterranean region. However, any such activities demand follow-up actions and resources (so-called “backbone funding”) to support their basic organizational operations, make concrete plans, assign roles and secure their sustainability [96,97].

## 4. Materials and Methods

To set the scene and enable the identification of background requirements to advance in marine biotechnology, a survey was set. In March 2021, selected experts—B-Blue project partners—from 8 Northern Mediterranean countries (Croatia, Greece, France, Italy, Montenegro, Portugal, Slovenia and Spain; 1–2 experts per project partner, 14 overall) were enrolled in a survey where respondents were asked to tick “five activities which you consider are most important for advancing marine biotechnology capability building in your country”. A total of 81 potential activities were suggested, originating from suggestions and solutions as outlined by the Blue Bioeconomy Forum [18]. The assumption behind this survey was that any activity that is considered important by several respondents points towards an important set of activities for the development of the marine biotechnology sector in the Northern Mediterranean region.

To identify good practices, we used a combination of desktop research and authors’ own expertise to collect and systematize the existing knowledge, allowing the identification of the local innovators. To benchmark the good practices in the Northern Mediterranean countries that participated in this project (Appendix A), each partner assembled the data on individual good practices, including: the organization behind the development of a best practice, funding source, short description of the idea, sector/field (aquaculture, cosmetics, health/pharmaceuticals, nutraceuticals, feed industry, energy, industrial processes, environment or other). Individual good practices were then categorized (business creation, technology transfer, technology support, funding mechanism, policy management, collaboration and networking, marketing/branding, other). The technological readiness and business levels were also determined.

To create national blue biotechnology hubs, we adopted a collaborative territorial approach to create an appropriate space for the interaction of communities with different knowledge and interests to reach a critical mass for the marine biotechnology sector, to generate innovation and transfer it to the commercialization stage. Initially, a stakeholder mapping exercise was performed where national country representatives from 8 Northern Mediterranean countries (Croatia, Greece, France, Italy, Montenegro, Portugal, Slovenia and Spain) mapped individual stakeholders based on their category (Administrative and public bodies, Industry and SMEs, Scientific institutions and academia, NGOs, Media representatives, Past and current projects). Afterwards, each stakeholder was assessed based on their interest and influence level in the field of marine biotechnology. This resulted in the creation of stakeholder maps. An example of a stakeholder map is provided in Figure 8. This exercise was performed to categorize and tailor the collaboration and engagement with stakeholders. The primary targets were those with high interest and high influence (top right quadrant on Figure 8). Stakeholders that are highly influential but have a low interest in the topic were targeted with activities to raise awareness, inform them and somehow incentivize them to grasp the importance of the topic. On the other hand, stakeholders with a high interest but low influence were used to inform them about our activities and provide a platform for their feedback. Finally, those with low interest and influence (bottom left quadrant) are monitored for their potential changes in either influence or interest in future. 

Five national blue biotechnology hubs were defined (France, Greece, Italy, Slovenia and Spain) through various activities. (1) National workshops (a minimum of one in each country) were organized with the aim of raising awareness on the marine biotechnology and blue bioeconomy sectors, summarizing the state-of-the-art, offering a platform for showcasing expertise by participants, for networking, discussing ideas related to the development of the sector and planning future activities and strategies. (2) Networking events such as work cafés were organized to exchange the opinions and needs of stakeholders, which enabled the definition of future steps towards implementing marine biotechnology value chains. (3) Technical workshops enabled an in-depth discussion on the selected value chains, the identification of potential bottlenecks and provided an opportunity to showcase the individual stakeholders’ expertise. (4) Hackathons provided a platform to find specific solutions and identify technological innovations in the maritime field, which can be applied to improve an existing situation or give rise to a new concept. (5) Strategic meetings provided a platform with fewer stakeholders that enabled the creating and further reinforcing of collaborative associations with the aim of defining participatory actions for further development of the selected value chains. They were also the tool of choice for communication between the five Northern Mediterranean blue biotechnology hubs (France, Greece, Italy, Slovenia and Spain). Throughout these events, the feedback from participants was used to identify bottlenecks from various stakeholder categories and is also included in the Discussion section.

## 5. Conclusions

This article assessed the current state of innovation in the marine biotechnology value chains development in the Northern Mediterranean. Focusing on eight Mediterranean countries (Croatia, Greece, France, Italy, Montenegro, Portugal, Slovenia and Spain), we benchmarked the current good practices to avoid the fragmentation of knowledge and prepare the national and international strategies. We have identified the sectors which are of highest potential for the Northern Mediterranean region. Algal production (macro- and microalgae) for added-value products, IMTA and the valorization of fisheries and aquaculture discards, bycatch and by-products could prepare the Northern Mediterranean countries to advance their innovation potential. Through the participatory activities we created the so-called blue biotechnology national hubs, i.e., ecosystems where collaborations, knowledge transfer and spill-over effects can occur to spur innovation and business within the marine biotechnology value chains. Considering that the establishment of a value chain is typically a long process that needs the involvement of technical and non-technical expertise, addressing the legislative framework (either national, regional or international one) with a severe reliance on the precarious nature of science and innovation financing, consolidated efforts are needed to push further the development of value chains with highest potential for development. Their success is not only dictated by the advances in research and development (i.e., technical skills) but, as we have shown, also by maintaining the non-technical operations. These include mutual learning platforms, monitoring, identification and sharing of good practices to build potential synergies, networking, advocacy and involvement with the policy making sector to iterate the legislation and guarantee a more stable funding and the further stimulation of innovation, which can benefit the whole Mediterranean economy. Additional activities should also build capacity regarding the market aspects and represent a proxy towards raising consumer awareness and acceptance. Therefore, special focus should be given to incentivize and finance the establishment of national blue biotechnology hubs with international outreach, where expertise, infrastructure and pilot sites can be shared to advance in the Northern Mediterranean sustainable bioeconomy.

## Figures and Tables

**Figure 1 marinedrugs-21-00416-f001:**
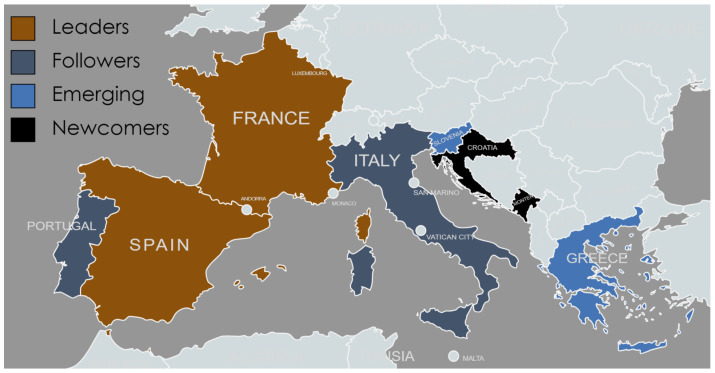
Northern Mediterranean countries (Croatia, Greece, France, Italy, Montenegro, Portugal, Slovenia and Spain) that participated in this study based on their development status (leader, follower, emerging and newcomer) as assessed in [3]. Created with mapchart.net.

**Figure 2 marinedrugs-21-00416-f002:**
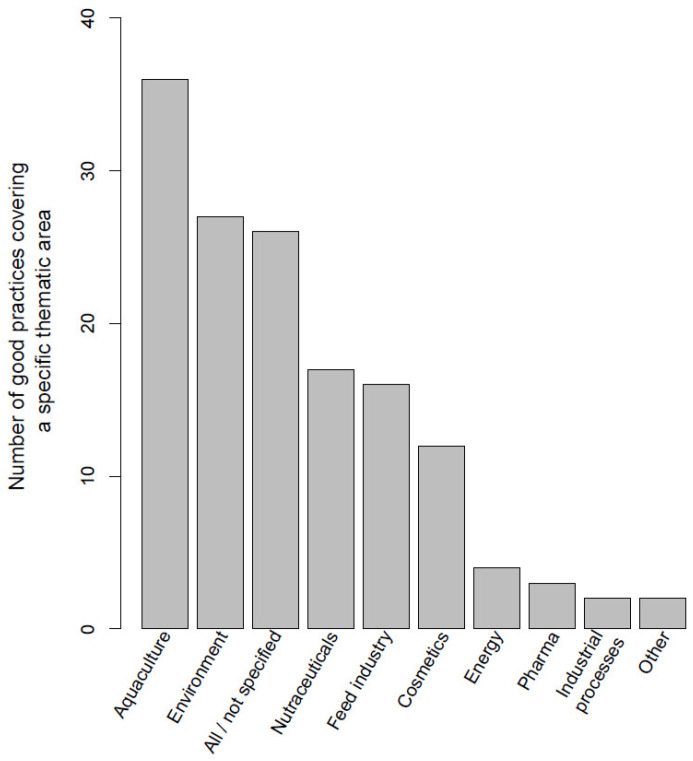
Number of good practice examples of marine biotechnology value chains that tackled a specific sector in the eight Northern Mediterranean countries (Croatia, Greece, France, Italy, Montenegro, Portugal, Slovenia and Spain). When a sector was not specified or when a good practice covered all/general marine biotechnology sectors, it was included in the “All/not specified group”. The energy sector mostly covers alternative sources of energy production, while the category “other” includes emerging sectors such as packaging, furniture or clothes. Note: most of these good practices involve the participation of international consortia with countries that are also outside the Mediterranean region. Nevertheless, at least one partner is from the aforementioned countries.

**Figure 3 marinedrugs-21-00416-f003:**
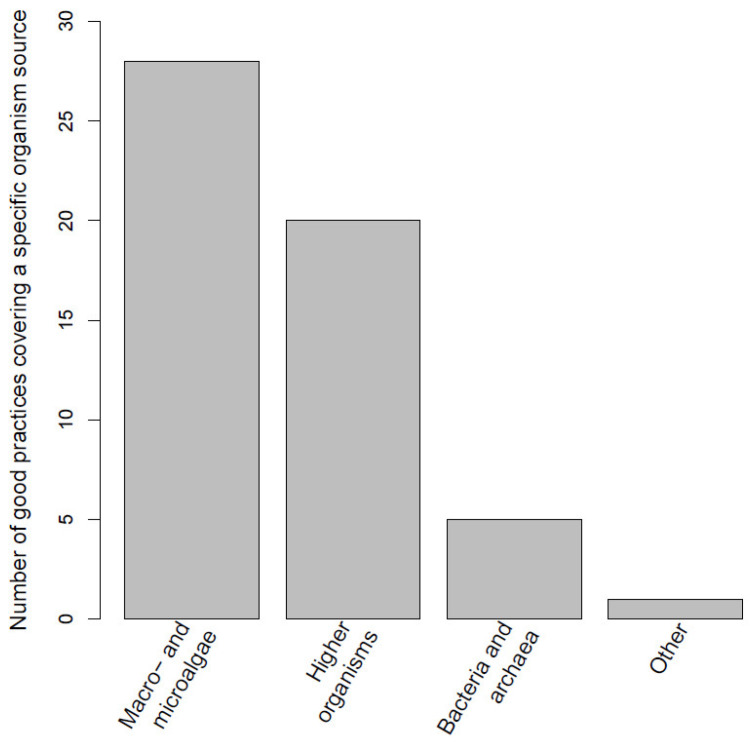
Number of good practices that target any of the four organism type categories (macro/micro algae, higher organisms—vertebrates, invertebrates and plants, bacteria and archaea, and others, such as viruses) in the assessed good practices among the existing marine biotechnology value chains that are developed within any of the eight Northern Mediterranean countries (Croatia, Greece, France, Italy, Montenegro, Portugal, Slovenia and Spain). Note: most of these good practices involve the participation of international consortia with countries that are also outside the Mediterranean region. Nevertheless, at least one partner is from the aforementioned countries.

**Figure 4 marinedrugs-21-00416-f004:**
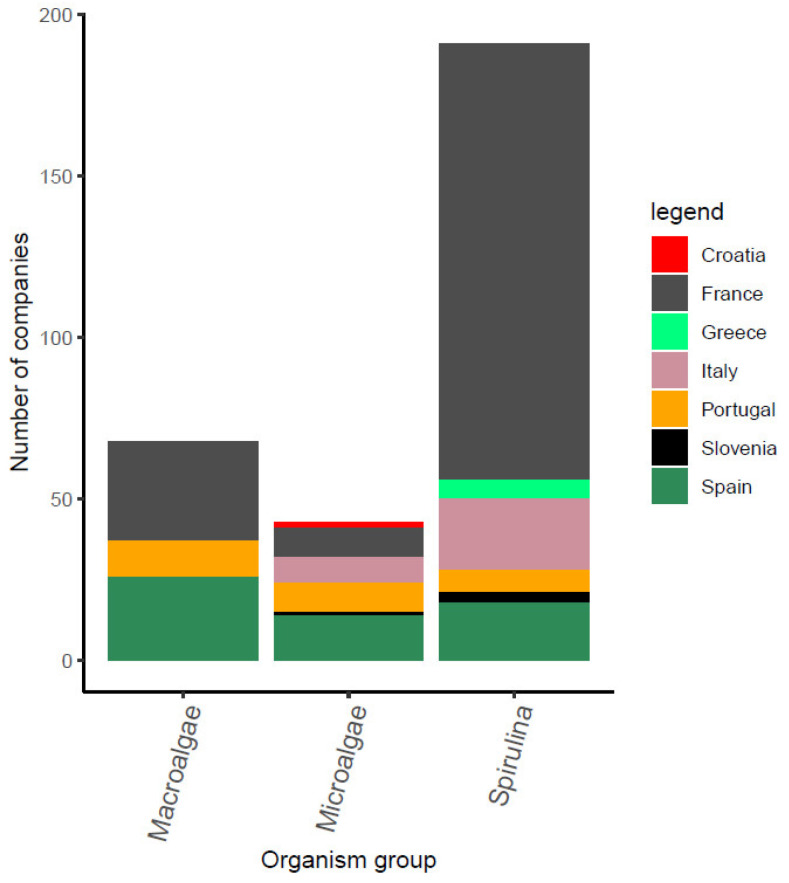
Registered companies in B-Blue countries that produce macroalgae, microalgae and *Spirulina*.

**Figure 5 marinedrugs-21-00416-f005:**
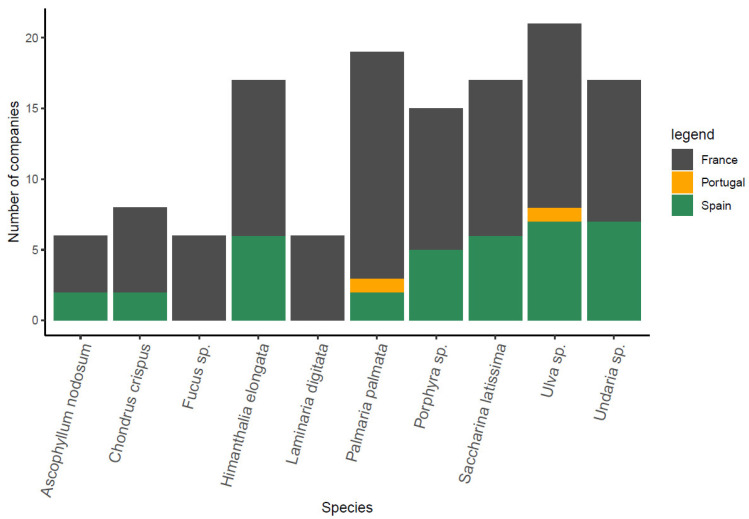
Macroalgal species that are most often cultivated in the B-Blue countries by registered companies. Different colors denote the producer countries.

**Figure 6 marinedrugs-21-00416-f006:**
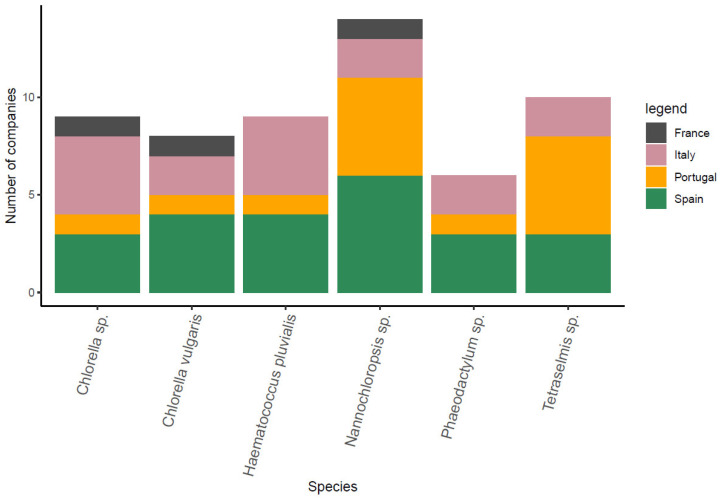
Microalgal species that are most often cultivated in the B-Blue partner countries by registered companies. Different colors denote the producer countries.

**Figure 7 marinedrugs-21-00416-f007:**
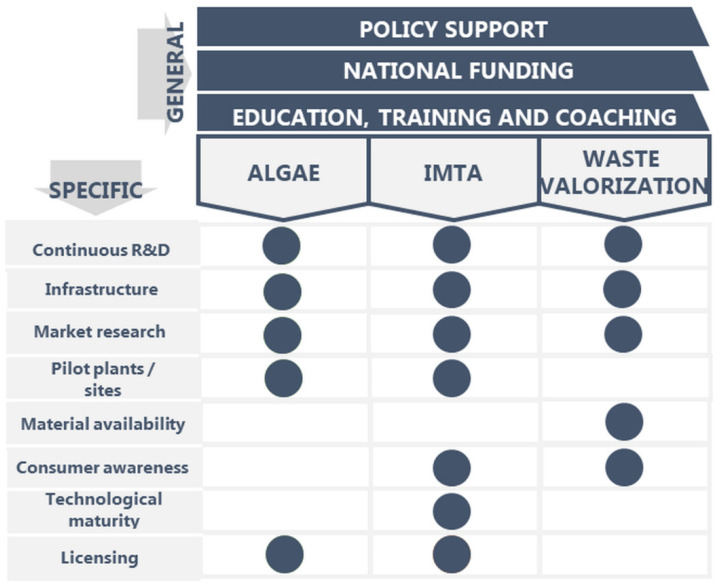
General bottlenecks to advance in marine biotechnology in the Northern Mediterranean region (top of the figure) and specific bottlenecks that were identified for three promising value chains for the region. The identification of bottlenecks was carried out as a combination of opinions among various stakeholders and project partners that were collected through desk research and direct interactions with experts.

**Figure 8 marinedrugs-21-00416-f008:**
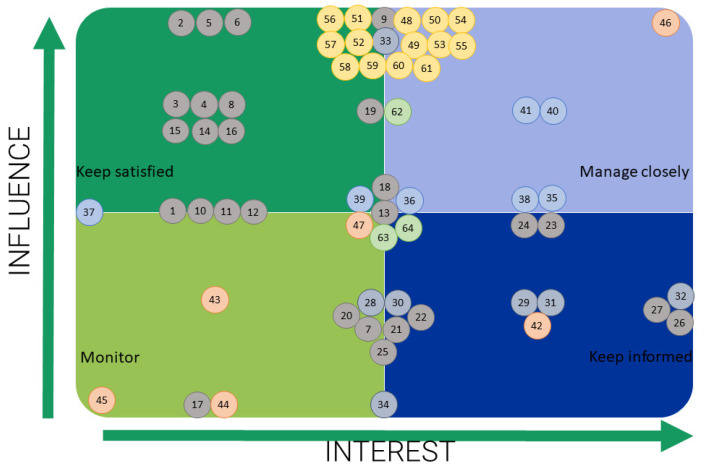
A stakeholder map example to identify the most relevant stakeholders for advancing the marine biotechnology field. The dots indicate individual stakeholders and they are numbered for their easier identification in the individual stakeholders’ lists (these are not presented due to confidentiality reasons), and their placement indicated their combined interest and influence in the sector. Based on their positioning on the map, these stakeholders can be invited to form strategic collaborations. Dots are color-coded to represent their sector (grey—administrative and public bodies; grey/blue—industry and SMEs; blue—scientific institutions and academia; orange—NGOs; yellow—media representatives; green—past and current projects).

**Table 1 marinedrugs-21-00416-t001:** Glossary of terms used in this manuscript.

Term	Description
Value chain	A value chain consists of a range of activities required to bring a product from its inception to its end consumer, through a series of steps involving physical transformation, input of various producer services and disposal after use [16]. Marine biotechnology value chains are tailored to various application sectors, all stemming from a generalized pipeline. It is composed of four steps: (i) basic research—bioprospecting, harvesting and collection of available biomass, either the whole organism, its parts or its associated microbiome. This is followed by (ii) applied research—collection/harvesting of biomass, preservation in culture collections or biobanks, cultivation and biomass processing, extraction, purification, structure elucidation and characterization of natural products, including laboratory scale applications to optimize the production conditions; (iii) industrial scale-up phase to sustain the production quantities; and (iv) commercial applications [14]. Different stakeholders from different organizations are typically involved in various stages of value chains and are faced with several challenges, especially the supply, technology development and definition of market needs [17].
Good practice	Existing knowledge that may refer to standards, regulations, methods, procedures, pilot actions and research results of innovative solutions that are applied and can be followed/transferred to build a resource-efficient society and promote the Sustainable Development Goals (SDGs). In addition, a good practice can be considered an implemented, ready-to-market project or/and a pilot action/research project with actual results. The full list of good practices in this study, covering the eight Northern Mediterranean countries, is included as Appendix A and the process of their identification is described in the Materials and Methods section.
Smart specialization	According to the EU and the Organization for Economic Co-operation and Development (OECD), it identifies a limited number of strategic areas for countries or regions therein that represent unique opportunities for boosting societal and economic development and growth. The strategic areas are identified by industrial/innovation, civil society and policy making stakeholders, using a combination of industrial, educational and innovation policies.

**Table 2 marinedrugs-21-00416-t002:** Average capture production statistics between 2014 and 2018 for eight Mediterranean countries. Data taken from the Food and Agriculture (FAO) Fisheries and Aquaculture statistics database. The discard rate was calculated based on published estimations for the Mediterranean Sea (average of 18.6%) [85].

Country	Average Capture Production 2010–2019 [tons]	Estimated Discard Rate (13.3–26.8%) [tons]
Croatia	Fish: 70,709	
	Mollusks: 1864	
	Crustaceans: 967	
	Total: 73,540	Total: 9781–19,709
France	Fish: 426,078	
	Mollusks: 74,750	
	Crustaceans: 15,885	
	Total: 516,714	Total: 68,723–138,479
Greece	Fish: 58,386	
	Mollusks: 6947	
	Crustaceans: 6182	
	Total: 71,515	Total: 9511–19,166
Italy	Fish: 134,790	
	Mollusks: 39,621	
	Crustaceans: 21,325	
	Total: 195,735	Total: 26,033–52,457
Montenegro	Fish: 1209	
	Mollusks: 158	
	Crustaceans: 34	
	Total: 1401	Total: 186–375
Portugal	Fish: 163,173	
	Mollusks: 17,597	
	Crustaceans: 1528	
	Total: 182,298	Total: 24,246–48,856
Slovenia	Fish: 288	
	Mollusks: 31	
	Crustaceans: 3	
	Total: 322	Total: 43–86
Spain	Fish: 897,491	
	Mollusks: 51,996	
	Crustaceans: 15,255	
	Total: 964,743	Total: 128,311–258,551
TOTAL for 8 Mediterranean countries	Fish: 1,752,125	
	Mollusks: 192,964	
	Crustaceans: 61,178	
	Total: 2,006,267	Total: 266,833–537,680

**Table 3 marinedrugs-21-00416-t003:** Average annual apparent seafood consumption between 2014 and 2018 and estimation of fish by-products category 3 from the processing of fish from fishery and aquaculture in the retail, commercial and processing chain. Data were calculated using the formula: *Average apparent seafood consumption = Fisheries production* (Table 1) *+ Aquaculture production + Imports − Exports* (Appendix A and tables therein). The by-product amounts were calculated based on conservative estimations for the Mediterranean region (15–30%).

Country	Average Yearly Apparent Seafood Consumption 2014–2018 [tons]	Estimated By-Products (15–30%) [tons]
Croatia	85,163	12,774–25,549
France	1,476,484	221,473–442,945
Greece	270,308	40,546–81,092
Italy	1,272,150	190,823–381,645
Montenegro	6075	911–1822
Portugal	429,799	64,670–128,940
Slovenia	14,852	2228–4455
Spain	1,795,766	269,365–538,730
TOTAL for 8 Mediterranean countries	5,350,596	802,589–1,605,179

## Data Availability

All data produced during this study consisted of interviews and desk research and are not included in this manuscript. They are presented as project deliverables on the project website (https://b-blue.interreg-med.eu/, accessed on 8 May 2023). In case some of the data are not publicly available, interested readers should address their inquiry to the corresponding author (A.R.).

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
