# Peer review of "Identification of Marine Biotechnology Value Chains with High Potential in the Northern Mediterranean Region"

_marinedrugs, 2023, doi:10.3390/md21070416_

Round 1

Reviewer 1 Report

ms Rotter et al.

This manuscript reports a state-of-art on the marine biotechnology development in some European countries around the Mediterranean Sea. The manuscript is developed as a position paper discussing on the different key-steps of the “value chains” and the challenges offered by the blue biotechnology in the Mediterranean Sea.

While the study points out many interesting aspects linked to the blue biotechnology research field development as well as to the transfer of knowledge and technology at an industrial scale, the manuscript is sometimes harsh to read.

General comments

. In the title and along the text, the manuscript emphasizes the “Mediterranean region”. I have some perplexities on this point, since the data reported and discussed by the authors come from 8 European countries (Spain, Portugal, France, Italy, Greece, Montenegro, Croatia and Slovenia) while the south Mediterranean Sea was not integrated in this study (e.g., Turky, Israel, Cyprus, Algeria, Tunisia, Maroc, Egypt, etc.).

I suggest therefore to modify the title and limit the study to the 8 European countries.

. The text is too long needing to be shortened in some parts and revised in its structure. Sometimes (e.g., section 3.2), the text needs to be revised, better separating the different aspects (e.g., macroalgae, microalgae).

. In its present state, this study looks like a project report more than a publication.

. The marine biotechnology related properties/peculiarities of the Mediterranean region are lacking. Why does the Mediterranean region important for marine biotechnology development? A paragraph on the factors influencing the biological resource production/use (e.g., biodiversity, biomass, temperature, salinity, light, etc.) would help the reader to better integrate the information and interests highlighted in this study.

. I suggest to add a map of the Mediterranean region reporting the different countries and the development of marine biotechnology field.

. I suggest to add a table for instance reporting the definition of the relevant terms and criteria, such as “value chains”, smart specialization strategies, good practices, etc.

Specific comments

-       Lines 38-39: I suggest removing “in the Mediterranean region” at the end of the sentence. The potential for innovation and growth is general for the blue biotechnology.

-       Line 118: associated microbiome? 

-       Line 118: “its parts”: fraction, extract, molecule

-       Line 117: I suggest remove “basic” from “basic steps”. 

-       Line 165: good practice?

-       Lines 165-173: good practices or best practices

-       Lines 164: 149 thematic areas: what are they? Reference?

-       Line 165: 55 good practices: what are they?

-       Figure 1: please, define better the fields: energy, other, industrial processes, all / not specified.

-       Figure 2: best practices? (good practices)

-       Fig.2: higher organisms: invertebrates, vertebrates, plants?

-       Fig.2: define “other”

-       Lines 183-200: seems more a report of B-Blue project than a publication.  What about higher education, university, etc.?

-       Lines 217-280: It is not clear to me what is “regional policies” vs European policies or national policies. This part might be shortened.

-       Section 3.1.2.: report of B-blue project?;  I suggest to discuss more or to remove. 

-       Lines 323 – 326: references of these two projects? 

-       Section 3.2.1.: separate macro- and micro-algae 

-       Line 384: remove. The taxonomic classification of Spirulina is cyanophyte. And cyanophytes are algae. The ref is almost old (2012).

-       Lines 415-428: Seaweed production in Spain and France is mainly developed in the Atlantic ocean coasts, not in the Med sea, mainly due to the tidal oscillations (high/low). This would open to discuss the challenges of seaweed production in the Med sea.

-       Lines 455-458: I suggest to replace the actual ref (48) with another more recent and representing better the mean of the sentence. 

-       Section materials and methods: is it really useful? Seems to be more a project report than a publication. 

-       Fig. 8: did the numbers in the circles included in the fig. 8 represent something?

Author Response

This manuscript reports a state-of-art on the marine biotechnology development in some European countries around the Mediterranean Sea. The manuscript is developed as a position paper discussing on the different key-steps of the “value chains” and the challenges offered by the blue biotechnology in the Mediterranean Sea.

While the study points out many interesting aspects linked to the blue biotechnology research field development as well as to the transfer of knowledge and technology at an industrial scale, the manuscript is sometimes harsh to read.

ANSWER: thank you for providing important feedback, which hopefully helped us to improve the manuscript. The individual comments were all addressed and the removed text is shown below and the modifications are also seen in the manuscript as tracked changes.

General comments

. In the title and along the text, the manuscript emphasizes the “Mediterranean region”. I have some perplexities on this point, since the data reported and discussed by the authors come from 8 European countries (Spain, Portugal, France, Italy, Greece, Montenegro, Croatia and Slovenia) while the south Mediterranean Sea was not integrated in this study (e.g., Turky, Israel, Cyprus, Algeria, Tunisia, Maroc, Egypt, etc.).

I suggest therefore to modify the title and limit the study to the 8 European countries.

Answer: thank you for pointing out this inconsistency and we fully agree that the results could have been much different for other countries, especially from the Southern Mediterranean. We have thus modified the title into: “The identification of marine biotechnology value chains with high potential for the Northern Mediterranean region”. The focus on eight countries is clearly introduced in the Abstract already and we also included the notation of Northern Mediterranean region elsewhere in the manuscript.

. The text is too long needing to be shortened in some parts and revised in its structure. Sometimes (e.g., section 3.2), the text needs to be revised, better separating the different aspects (e.g., macroalgae, microalgae).

ANSWER: Thank you for this feedback. Section 3.2.1 was further separated to 3.2.1.1 (Macroalgae cultivation) and 3.2.1.2 (Microalgae cultivation) to enable an easier division of the text for the reader.

A part of the text from the beginning of Section 3.2.1 concerning only macroalgae was moved to the beginning of 3.2.1.1: “The global production of macroalgae amounts up to 35 million tons (Mt) of fresh weight annually (where 97% of its biomass derives from aquaculture and the rest is harvested) and around 24% and less than 1% of it is contributed within the EU for harvesting and aquaculture, respectively [36]. The global production is still primarily dominated by two Asian countries, namely China and Indonesia, producing together >90% of the global market supply [37]. Hence, although Europe production is currently small-scale, the macroalgal sector is considered as the most notable subsector in Euro-pean blue bioeconomy and the projections imply an expansion in European annual production from around 0.3 Mt up to 8 Mt by 2030 which could create up to 85,000 jobs [28,38]. It is estimated that the key seaweeds grown in the Mediterranean countries (including their Atlantic coasts) in 2030 could include for instance sea lettuce (Ulva lactuca) for human consumption, sugar kelp (Saccharina latissima) for use in food prod-ucts and animal feed, dulse (Palmaria palmata) for the food and cosmetics sectors, Asparagopsis taxiformis for cattle feed additives with methane-reducing properties, and oarweed (Laminaria digitata) to produce alginate for use in the food additives and biopackaging segments [38]. An emerging field is also the seaweed farming for ecological restoration of marine macroalgal forests, given the predicted upscaling from small-scale, short-term academic experiments to industry and restoration practitioners, required to secure the oceans’ sustainable future [39] and building up in the Mediterranean region on the knowledge produced by successful projects such as AFRIMED (http://afrimed-project.eu/) and ROCPOP-life (http://www.rocpoplife.eu/). Regardless on the target value chain, however, it is important to use the basic ecological knowledge of these species and avoid their introduction into new areas, especially where they might be considered invasive, such as is the case for some Asparagopsis species [40].”

In addition, text was shortened in several parts, which hopefully makes the whole manuscript more concise. The deleted segments are striked through as shown below:

  • To develop the most promising marine biotechnology value chains for the Northen Mediterranean region, project partners, who are internationally renowned experts in either (i) marine biotechnology domain, (ii) provide non-technical skills or (iii) both, conducted the work in a three-step process: analyze-transfer-capitalize.
  • More specifically, the three identified requirements that received most votes from the experts and need to be addressed to enable the launch of the value chains were legislation and policy support, financing, collaboration through knowledge creation. : (i) Highlight marine biotechnology in Smart Specialization Strategies; (ii) Increase funding at national level to support the continuation of projects after their first financing round; and (iii) Bridge the collaboration gap between research and industry by education, training and coaching.
  • Deleted the second sentence “This exercise enabled to highlight the innovation potential in the marine biotechnology field and define the value chains that are of strategic importance to capitalize the needs and expertise in the Mediterranean countries, as well as those with high market potential.” from the beginning of Section 2.2
  • In the 3.2.2 paragraph on IMTA, a redundant part was deleted and is not available in the revised version. To avoid the confusion among the readers, we left in the text only the concept of industrial symbiosis within IMTA and excluded the redundant part: “An interesting fact also emerges in the IMTA potential in the Northern Mediterranean region. We argued in the beginning of the article that one of the main bottlenecks of advancing in marine biotechnology is the fragmentation of knowledge and businesses. To advance in the technology readiness of any of the selected value chains, there is therefore a need for transdisciplinary and transnational collaborations. This is true when establishing IMTA practices, but only initially. Initially, there is a need to ensure the integration between the separate companies at an operational level in order to deal with different production cycles, processing the different components of the IMTA system and the availability of infrastructure. The next level demands the fragmentation of activities to find markets and distribution networks for the additional extractive organisms. This is well encompassed in the context of industrial symbiosis,
  • We deleted the last paragraph of section 3.3, as well as the former Figure 7 as after consideration, we believe this deletion does not impact the overall essence of the manuscript: “Ideally, we would propose a form of sustainability or formalization of these national blue biotechnology hubs through securing national and international funds or capitalizing them in forms of spin offs or creation of professional clusters. When these clusters are integrated into new or existing international ones, it is however important to maintain their regional independence to some extent as regionalization of innovation policy more accurately considers the regional specific context and circumstances in terms of the industrial structure, institutional set-up and knowledge base [89]. Such blue biotechnology hubs should be of strategic importance for the Mediterranean region to enable a faster adoption of marine biotechnology and facilitate stakeholders towards a faster development and application of promising value chains without the need to duplicate the efforts and reinvent the wheel for processes which were already initiated but had no possibility to display their good practices or span beyond national borders due to a lack of established blue biotechnology hubs. With the aim of having these hubs to act as national and regional innovation catalysts, we propose a roadmap for development of new marine biotechnology value chains in the Mediterranean region (Figure 7). We propose the process in five steps, where the first four are essential prerequisites for success and should be implemented even before the actual technical part (in the laboratories or on pilot sites) – step 5. These are the (i) contextual steps (step 1 – benchmark the current sets of good practices that will also enable the identification of key stakeholders and determine possible bottlenecks that limit innovation and the development of marine biotechnology value chains; and step 3 – prepare action plans with newly established collaborative networks by defining roles, responsibilities, main activities and -especially- the main gains in participating in the process) and (ii) strategic steps (step 2 – building the community by implementation of activities that enable the co-creation of knowledge and step 4 – advocacy with important stakeholders to provide financial and legislative support).”
  • A very long sentence in the conclusion section was shortened: “These include mutual learning platforms, monitoring, identification and sharing of good practices to build potential synergies, networking, advocacy and involvement with the policy making sector to iterate the legislation and guarantee a more stable funding and further stimulate innovation, better collaboration between the public and private sector stakeholders to stimulate investment and development of value chains, securing funding for maintenance of national blue biotechnology hubs, transfer of knowledge and provision of experts to span beyond the national borders and which can benefit the whole Mediterranean economy.”
  • The last sentence in the conclusion was shortened: “Therefore, special focus should be given to incentivize and finance the establishment of national blue biotechnology hubs with an international outreach, where expertise, infrastructure and pilot sites can be shared to advance in the Northern Mediterranean sustainable bioeconomy, thus enabling transformational change, engaging Mediterra-nean society, making decisions circular and environmentally friendly.”

. In its present state, this study looks like a project report more than a publication.

ANSWER: Thank you for pointing this out. We assure the reviewer that the manuscript is definitely not a project report, and we hopefully do not give this impression after the corrections. However, without having obtained the project, a lot of research and feedback and detailed country assessment would have never been possible. Hence, we feel it is important to acknowledge the importance of the project. We have nevertheless omitted specific wording that might leave an impression of reading a project report.

  • In section 2.3 we omitted the mentioning of B-Blue: “In B-Blue project wWe shall call them “Blue Biotechnology Hubs (BBt Hubs)”.
  • In the section 3.2.4, we have shortened the introductory part and omitted the mentioning of B-Blue to avoid sounding as if this was a project report: “Importantly, the list of the specific bottlenecks is a living document, drafted as a result of direct interactions by B-Blue stakeholders. The bottlenecks might change over time, depending on the financial, research, regulatory and time efforts spent to develop the three value chains.”
  • For the same reason, the Title of Figure 7 is slightly shortened: “…The identification of bottlenecks was done as a combination of project expert opinions among various stakeholders and project partners that were collected through desk research and direct interactions with experts (to define the general bottlenecks – in the step 1: analysis/setting the scene part of the B-Blue project activities) and expert stakeholder opinions that were collected during project activities (also described in the Materials and methods section).”
  • In the 3.2.4 section, the part speaking about IMTA, the corrected sentence now reads as: “In the Northern Mediterranean region that is heavily dependent on tourism, aware-ness raising campaigns should target the potential consumers and industry to under-stand environmental benefits and eliminate the stigma of IMTA being a “waste of coastal space”, as was mentioned during the events organized by B-Blue partnersobserved by the stakeholders.”

. The marine biotechnology related properties/peculiarities of the Mediterranean region are lacking. Why does the Mediterranean region important for marine biotechnology development? A paragraph on the factors influencing the biological resource production/use (e.g., biodiversity, biomass, temperature, salinity, light, etc.) would help the reader to better integrate the information and interests highlighted in this study.

ANSWER: Thank you for pointing this out. The general description of the Mediterranean Basin was improved in the Introduction section and it now reads as: “The Mediterranean Basin, located across the South of Europe and the North of Africa, is unique by virtue of its history, cultural heritage climate, diet, ecosystems, and being a global hotspot of biological diversity (with over 17,000 species) with a high rate of endemism [7–9]. The Mediterranean Basin is mostly oligotrophic and its biological productivity decreases from north to south and west to east, while the opposite trend is observed for salinity and temperature [10]. In theory, the Mediterranean thus offers natural and societal opportunities to valorize the rich biodiversity and advance in the marine biotechnology and the blue bioeconomy sectors.”

A new reference was included [10].

. I suggest to add a map of the Mediterranean region reporting the different countries and the development of marine biotechnology field.

ANSWER: Thank you for this useful suggestion. We have categorized the participating countries into 4 groups based on the criteria that are explained in the newly added text:

“These eight countries differ in their maturity level of marine biotechnology, which was recently assessed in terms of these indicators: (i) aquatic macroorganisms aquaculture, macro- and microalgal aquaculture, (ii) existence of marine (blue) biotechnology in national and regional Smart Specialization Strategies (Table 1), and (iii) obtained funding for marine biotechnology projects and support measures [ref ]. By combining these indicators, four groups are represented this study: Mediterranean leaders (France and Spain, with implemented innovation strategies, national/regional legislation and existing industry), followers (Italy and Portugal), with existing legislative and financial support but several value chains – see Table 1 – that were not fully finalized), emerging countries (Greece and Slovenia), with some legislative support but a currently lower financial support and less established aquaculture sector, and newcomers (Croatia and Montenegro), with established commitment on scientific or legislative side but lacking concrete implemented innovations.”

The newly created Figure 1 is also included now in the manuscript, along with its caption: “Northern Mediterranean countries (Croatia, Greece, France, Italy, Montenegro, Portugal, Slovenia and Spain) that participated in this study based on their development status (leader, follower, emerging and newcomer) as assessed in [3]. Created with mapchart.net.”

Please also note that the referred Table 1 is newly created as well to address your comment below.

. I suggest to add a table for instance reporting the definition of the relevant terms and criteria, such as “value chains”, smart specialization strategies, good practices, etc.

ANSWER: Thank you for this suggestion. We have created the new Table 1 (please see in manuscript) with introduced terminology that was copied here from the text in the article (thus also reducing the article length): value chain, smart specialization, good practice.

Specific comments

-       Lines 38-39: I suggest removing “in the Mediterranean region” at the end of the sentence. The potential for innovation and growth is general for the blue biotechnology.

ANSWER: Done.

-       Line 118: associated microbiome? 

ANSWER: Done, thank you. This part now constitutes Table 1 and its definition of a value chain.

-       Line 118: “its parts”: fraction, extract, molecule

ANSWER: Done, thank you. This part now constitutes Table 1 and its definition of a value chain and reads as: …” the whole organism, its parts (e.g., fractions, extracts, molecules) or its associated microbiome…”

-       Line 117: I suggest remove “basic” from “basic steps”. 

ANSWER: done, thank you.

-       Line 165: good practice?

ANSWER: former line 165 had good practices already in text, but we have noticed, however, that the y-axis in Figure 2 (former Figure 1) had written “best practices” instead of good practices. We have corrected this error and have uploaded the corrected Figure in this submission version.

-       Lines 165-173: good practices or best practices

ANSWER: Thank you for this observation. As above, we corrected the error on the y-axis in Figure 3 (old Figure 2) as well.

-       Lines 164: 149 thematic areas: what are they? Reference?

ANSWER: Thank you for raising this important inconsistency. We now realize that our own naming “thematic areas” was misleading. Therefore, we have omitted the use of this term, which just described the overall sum of all sectors in the list of good practices. We have therefore corrected the text to avoid the misleading use of the term “thematic areas”, which reads now “These individual good practices focused on one, more or all sectors (aquaculture, cosmetics, energy, environment, feed industry, industrial processes – enzyme and catalysts, nutraceuticals, pharmaceuticals, other) and are shown in Figure 2.” Also, in the process we also uncovered a duplicated entry in the list of good practices. Therefore, we have also submitted a new Supplementary Table 1 without the duplicated entry. Hence, in the improved submitted version there are now 89 good practices and not 90, which was corrected in text as well.

-       Line 165: 55 good practices: what are they?

ANSWER: The process of establishing good practices is now introduced in Table 1.

-       Figure 1: please, define better the fields: energy, other, industrial processes, all / not specified.

ANSWER: This was defined and the title of Figure 2 (previously Figure 1) is now: “Figure 2. Number of good practice examples of marine biotechnology value chains that tackled a specific sector in the eight Northern Mediterranean countries (Croatia, Greece, France, Italy, Montenegro, Portugal, Slovenia and Spain). When a sector was not specified or when a good practice covered all/general marine biotechnology sectors, it is included in the “All/not specified group”. The energy sector mostly covers alternative sources of energy production, while the category “other” includes emerging sectors such as packaging, furniture or clothes. Note: most of these good practices involve the participation of international consortia with countries that are also outside the Mediterranean region. Nevertheless, at least one partner is from the aforementioned countries.”

-       Figure 2: best practices? (good practices)

ANSWER: Thank you very much for this observation. As stated above, the y-axes of both Figure 2&3 (formerly Figure 1&2) have now been corrected (best practices changed to good practices).

-       Fig.2: higher organisms: invertebrates, vertebrates, plants?

ANSWER: Figure 3 (former Figure 2) legend was improved by introducing each of the four categories … “Figure 3. Number of good practices that target any of the four organism type categories (mac-ro/micro algae, higher organisms – vertebrates, invertebrates and plants, bacteria and archaea, and others, such as viruses) in the assessed good practices…”

-       Fig.2: define “other”

ANSWER: Same answer as above.

-       Lines 183-200: seems more a report of B-Blue project than a publication.  What about higher education, university, etc.?

ANSWER: Thank you for this observation. After having conducted stakeholder analysis and mapping in the past and working, we feel that there is lack of understanding of its importance and the effort needed. We therefore wanted to take this opportunity to introduce the stakeholder mapping concept (in Materials and Methods) and the output – over 600 engaged stakeholders after the mapping and around 1,500 people who engaged in our activities is a good critical mass to really provide significant input and feedback. We therefore believe this is not merely a reported number but it can help grasp the reader on the importance. Nevertheless, we included some more explanation and refer the reader to M&M, while the academic stakeholders were already included in the originally submitted version: “To capitalize the existing knowledge and prevent its loss that typically occurs after the end of financing rounds, we first conducted a stakeholder mapping exercise (see Materials and Methods on how to conduct and map the stakeholders, which optimizes the engagement effort to yield new collaborations, funding or legislation change). 636 potentially interested stakeholders from several sectors and activities (administration, research and academia”…

-       Lines 217-280: It is not clear to me what is “regional policies” vs European policies or national policies. This part might be shortened. 

 ANSWER: Thank you for this feedback. Regional policies are those policies regulating marine biotechnology at subnational level, national policies are those of the single member states, while EU policies are those policies at European level. In our opinion, this is one of the most significant parts of the manuscript. The reasons are threefold: (1) policy support was considered by our experts as the essential prerequisite to advance in the field of marine biotechnology, (2) it is extremely difficult to find information on policy in relation with marine biotechnology, but (3) the policy-making sector and financers of innovative activities are demanding policy-related outputs in every funded activity. Therefore, we believe this part should stay and it well embraces the European, regional (Euro-Mediterranean) and of course national policies. We however agree that the term “regional” was misleading, hence we put the better wording “trans-regional” instead.

-       Section 3.1.2.: report of B-blue project?;  I suggest to discuss more or to remove.

 ANSWER: Please refer to the answer above. These words were to address the problematic that, often, due to the lack of funding (and monitoring mechanisms!) We have therefore added some text to the existing one: “Ideally, national funding would critically review the capitalization potential of the concluded activities and only those with realistic prospects to further increase their technological readiness would be financed in the next rounds, thus preventing the loss of knowledge and results generated during the financing rounds.”

Based on your suggestion, we have nevertheless excluded the lengthy explanation of two national cases in order not to divert the reader from the most important thought in this Section and this text does not appear in this submitted version: “Existing good practice examples include the Innovation in the fisheries and aquacul-ture service, a service offered by the Ministry of Agriculture, Fisheries and Food of the government of Spain, with public access to relevant information related to innovation in the fisheries and aquaculture sector such as R&D strategic plan, map of technologi-cal platforms and capacity maps. Additionally, the Spanish PLEAMAR Programme of the Ministry of Ecological Transition and Demographic Challenge and implemented through the Biodiversity Foundation aims to support the fishing and aquaculture sec-tor through annual grants in its commitment to sustainability and to the protection and conservation of biodiversity and natural heritage.”

-       Lines 323 – 326: references of these two projects?

 ANSWER: Thank you for this, the references to these two projects are now included in the reference list.

-       Section 3.2.1.: separate macro- and micro-algae

ANSWER: Thank you for this suggestion, two sub-sections were added -  3.2.1.1 for macroalgal and 3.2.1.2 for microalgal production.

-       Line 384: remove. The taxonomic classification of Spirulina is cyanophyte. And cyanophytes are algae. The ref is almost old (2012)

 ANSWER: To avoid further confusion among the readers we amended the sentence using some newer publications and also followed the logic behind the important study of Araujo et al. (2021), that is also cited. The new sentence now reads as: “The taxonomic classification of Arthrospira (Spirulina) either into cyanobacteria or algae is an ongoing discussion in the scientific community [33,34] and is beyond the scope of this work, hence, it is represented as a standalone taxon which is also historically very important from the consumer’s and industrial perspective [34,35].”

The new citations that were referenced here are also in the reference list: (Araujo et al., 2021, Front. Mar. Sci; Nowicka-Krawczyk et al., 2019, Sci Reports)

-       Lines 415-428: Seaweed production in Spain and France is mainly developed in the Atlantic ocean coasts, not in the Med sea, mainly due to the tidal oscillations (high/low). This would open to discuss the challenges of seaweed production in the Med sea.

 ANSWER: This is a very important consideration and we have included a new sentence (the new sentence is highlighted in yellow), outlining the main challenges of seaweed production (with 3 new references):

Seaweed production (both harvesting from wild stocks and aquaculture) is primarily concentrated in the Atlantic region with few units cultivating species that are native in the Mediterranean Sea e.g., Ulva sp., Gracilaria sp. [39]. The main reasons behind a slower adoption of macroalgal cultivation in the Mediterranean waters are the need to (i) target native Mediterranean species with adapted cultivation techniques for specific sectors/purposes/locations/species, (ii) create germplasm banks to ensure the preservation of desirable local traits and genetic diversity, (iii) ensure the availability of suitable cultivation sites, but also (iv) lack of investment, and (v) the restrictive and inflexible implementation of European environ-mental regulations on aquaculture [46-48].

-       Lines 455-458: I suggest to replace the actual ref (48) with another more recent and representing better the mean of the sentence.

 ANSWER: Thank you for this useful suggestion. We have corrected the sentence overall into: “The microalgal biotechnology industry is of high interest due to their pigments, carbohydrates, lipids and proteins productivity potential that can be used as food and feed supplements and a broad variety of other applications, such as pharmaceuticals, cosmetics or agriculture ones  availability of multiple high-value products such as pigments, carbohydrates, proteins, nutraceuticals, biopharmaceuticals for a broad variety of applications [55,56]”. A newer reference (Barbosa et al., 2023, Trends in Biotechnology) was added as well.

-       Section materials and methods: is it really useful? Seems to be more a project report than a publication.

 ANSWER: We followed the logical structure as instructed by the Journal itself. Also, some of the methods (such as stakeholder mapping, finding good practices) are significantly consisting of desk research, which is often undermined when constructing strategies that will enable future financing rounds. For this reasons we believe it is useful to leave this section as a lot of research hours were put into the work presented.

-       Fig. 8: did the numbers in the circles included in the fig. 8 represent something?

ANSWER: Thank you for the very observant comment! Indeed, the numbers represent individual stakeholders. For clarification and to avoid suggesting the reader that their lists are provided, we added this text into the Figure caption: …” The dots indicate individual stakeholders and they are numbered for their easier identification in the individual stakeholders’ lists (these are not presented due to confidentiality reasons), and their placement indicated their combined interest and influence in the sector.”…

Reviewer 2 Report

This is a very insightful paper giving readers a broad and at the same time detailed idea of the state-of-the-art of innvation in marine biotechnology in selected European countries. The manuscript is generally well-written, it has a logical structure although it seems to be too lengthy. I did not spotted any issues, only two notes:

- Did the authors plainly define "best practices" as "exisiting knolwledge"? In my view it is too vague, clear criteria should be provided.

- The green color (or even all colors) in Fig. 6 are misleading in combination with the cross sign. I'd suggest eliminating the colors in this figure.

Author Response

This is a very insightful paper giving readers a broad and at the same time detailed idea of the state-of-the-art of innvation in marine biotechnology in selected European countries. The manuscript is generally well-written, it has a logical structure although it seems to be too lengthy. I did not spotted any issues, only two notes:

ANSWER: Thank you for providing your feedback. The article was shortened in many parts, which hopefully makes the whole manuscript more concise. The deleted segments are striked through as shown below and all changes are tracked in the revised submission:

  • To develop the most promising marine biotechnology value chains for the Northen Mediterranean region, project partners, who are internationally renowned experts in either (i) marine biotechnology domain, (ii) provide non-technical skills or (iii) both, conducted the work in a three-step process: analyze-transfer-capitalize.
  • More specifically, the three identified requirements that received most votes from the experts and need to be addressed to enable the launch of the value chains were legislation and policy support, financing, collaboration through knowledge creation. : (i) Highlight marine biotechnology in Smart Specialization Strategies; (ii) Increase funding at national level to support the continuation of projects after their first financing round; and (iii) Bridge the collaboration gap between research and industry by education, training and coaching.
  • Deleted the second sentence “This exercise enabled to highlight the innovation potential in the marine biotechnology field and define the value chains that are of strategic importance to capitalize the needs and expertise in the Mediterranean countries, as well as those with high market potential.” from the beginning of Section 2.2
  • In the 3.2.2 paragraph on IMTA, a redundant part was deleted and is not available in the revised version. To avoid the confusion among the readers, we left in the text only the concept of industrial symbiosis within IMTA and excluded the redundant part: “An interesting fact also emerges in the IMTA potential in the Northern Mediterranean region. We argued in the beginning of the article that one of the main bottlenecks of advancing in marine biotechnology is the fragmentation of knowledge and businesses. To advance in the technology readiness of any of the selected value chains, there is therefore a need for transdisciplinary and transnational collaborations. This is true when establishing IMTA practices, but only initially. Initially, there is a need to ensure the integration between the separate companies at an operational level in order to deal with different production cycles, processing the different components of the IMTA system and the availability of infrastructure. The next level demands the fragmentation of activities to find markets and distribution networks for the additional extractive organisms. This is well encompassed in the context of industrial symbiosis,
  • We deleted the last paragraph of section 3.3, as well as the former Figure 7 as after consideration, we believe this deletion does not impact the overall essence of the manuscript: “Ideally, we would propose a form of sustainability or formalization of these national blue biotechnology hubs through securing national and international funds or capitalizing them in forms of spin offs or creation of professional clusters. When these clusters are integrated into new or existing international ones, it is however important to maintain their regional independence to some extent as regionalization of innovation policy more accurately considers the regional specific context and circumstances in terms of the industrial structure, institutional set-up and knowledge base [89]. Such blue biotechnology hubs should be of strategic importance for the Mediterranean region to enable a faster adoption of marine biotechnology and facilitate stakeholders towards a faster development and application of promising value chains without the need to duplicate the efforts and reinvent the wheel for processes which were already initiated but had no possibility to display their good practices or span beyond national borders due to a lack of established blue biotechnology hubs. With the aim of having these hubs to act as national and regional innovation catalysts, we propose a roadmap for development of new marine biotechnology value chains in the Mediterranean region (Figure 7). We propose the process in five steps, where the first four are essential prerequisites for success and should be implemented even before the actual technical part (in the laboratories or on pilot sites) – step 5. These are the (i) contextual steps (step 1 – benchmark the current sets of good practices that will also enable the identification of key stakeholders and determine possible bottlenecks that limit innovation and the development of marine biotechnology value chains; and step 3 – prepare action plans with newly established collaborative networks by defining roles, responsibilities, main activities and -especially- the main gains in participating in the process) and (ii) strategic steps (step 2 – building the community by implementation of activities that enable the co-creation of knowledge and step 4 – advocacy with important stakeholders to provide financial and legislative support).”
  • A very long sentence in the conclusion section was shortened: “These include mutual learning platforms, monitoring, identification and sharing of good practices to build potential synergies, networking, advocacy and involvement with the policy making sector to iterate the legislation and guarantee a more stable funding and further stimulate innovation, better collaboration between the public and private sector stakeholders to stimulate investment and development of value chains, securing funding for maintenance of national blue biotechnology hubs, transfer of knowledge and provision of experts to span beyond the national borders and which can benefit the whole Mediterranean economy.”
  • The last sentence in the conclusion was shortened: “Therefore, special focus should be given to incentivize and finance the establishment of national blue biotechnology hubs with an international outreach, where expertise, infrastructure and pilot sites can be shared to advance in the Northern Mediterranean sustainable bioeconomy, thus enabling transformational change, engaging Mediterra-nean society, making decisions circular and environmentally friendly.”

- Did the authors plainly define "best practices" as "exisiting knolwledge"? In my view it is too vague, clear criteria should be provided.

ANSWER: Thank you for this comment. We believe that the best practices can be identified only after a thorough peer review of the good practices to assess their capitalization potential and transferrability. For this reason, we decided to stick to the term »good practices« The approach of their identification is described in the Materials and methods section: “To identify good practices, we used a combination of desktop research and authors’ own expertise to collect and systematize the existing knowledge, allowing to identify the local innovators. To benchmark the good practices in the Northern Mediterranean countries that participated in this project (Supplementary Table 1), each partner assembled the data on individual good practices, including: the organization behind the development of a best practice, funding source, short description of the idea, sector/field (aquaculture, cosmetics, health/pharmaceuticals, nutraceuticals, feed industry, energy, industrial processes, environment or other). Individual good practices were then categorized (business creation, technology transfer, technology support, funding mechanism, policy management, collaboration and networking, marketing/branding, other). The technology readiness and business levels were also determined.«

We also newly added Table 1, which now includes the description of the terminology used in the manuscript (thus further shortening it). » Existing knowledge that may refer to standards, regulations, methods, and procedures that are applied and can be followed/transferred to build a resource-efficient society and promote the Sustainable Development Goals (SDGs). In addition, a good practice can be considered an implemented, ready to market project or/and a pilot action/research project with actual results. The full list of good practices in this study, covering the eight Northern Mediterranean countries, is included as Supplementary Table 1 and the process of their identification is described in the Materials and Methods section”.

- The green color (or even all colors) in Fig. 6 are misleading in combination with the cross sign. I'd suggest eliminating the colors in this figure.

ANSWER: Thank you for this important observation. We agree and have removed the cross, as well as used only one colour. The improved Fig. 6 (now Figure 7) is now included in the manuscript.

Round 2

Reviewer 1 Report

the authors revised the manuscript according to the reviewers'reports

I don't have other comments